# MAPK/ERK Signaling in Regulation of Renal Differentiation

**DOI:** 10.3390/ijms20071779

**Published:** 2019-04-10

**Authors:** Kristen Kurtzeborn, Hyuk Nam Kwon, Satu Kuure

**Affiliations:** 1Helsinki Institute of Life Science, University of Helsinki, FIN-00014 Helsinki, Finland; kristen.kurtzeborn@helsinki.fi (K.K.); hyuk.kwon@helsinki.fi (H.N.K.); 2Stem Cells and Metabolism Research Program, Faculty of Medicine, University of Helsinki, FIN-00014 Helsinki, Finland; 3GM-Unit, Laboratory Animal Center, Helsinki Institute of Life Science, University of Helsinki, FIN-00014 Helsinki, Finland

**Keywords:** extracellular signal-regulated kinase, MAPK/ERK signaling, intracellular signaling, kidney development, ureteric bud branching morphogenesis, nephrogenesis, progenitor cells, self-renewal, differentiation

## Abstract

Congenital anomalies of the kidney and urinary tract (CAKUT) are common birth defects derived from abnormalities in renal differentiation during embryogenesis. CAKUT is the major cause of end-stage renal disease and chronic kidney diseases in children, but its genetic causes remain largely unresolved. Here we discuss advances in the understanding of how mitogen-activated protein kinase/extracellular signal-regulated kinase (MAPK/ERK) activity contributes to the regulation of ureteric bud branching morphogenesis, which dictates the final size, shape, and nephron number of the kidney. Recent studies also demonstrate that the MAPK/ERK pathway is directly involved in nephrogenesis, regulating both the maintenance and differentiation of the nephrogenic mesenchyme. Interestingly, aberrant MAPK/ERK signaling is linked to many cancers, and recent studies suggest it also plays a role in the most common pediatric renal cancer, Wilms’ tumor.

## 1. Introduction

The developing kidneys undergo complex morphogenesis to become fully functional, non-regenerating organs at birth. Renal morphogenesis is guided by classical inductive interactions, which are now known to take place through cell-cell-induced signaling, leading to activation of cell-intrinsic intracellular cascades. It has become evident that receptor tyrosine kinase (RTK)-activated signaling is a key regulator of kidney development [1,2]. The major inducers of RTK signaling in the early developing kidney include glial cell line-derived neurotrophic factor (GDNF)-induced RET signaling and fibroblast growth factor (FGF) signaling. Binding of these growth factors to their cognate receptors causes receptor dimerization, autophosphorylation, and activation of intracellular signaling cascades, including the mitogen-activated protein kinase/extracellular signal-regulated kinase (MAPK/ERK) pathway [3,4]. The significant roles of the MAPK/ERK pathway in morphogenesis are emerging [5,6,7,8], and we recently reported the activation of MAPK/ERK patterns and functions in nephron progenitors (NPs) of the developing kidney [9]. However, the full spectrum of MAPK/ERK functions at the cellular level, and specifically during renal development, are only partially determined. This review focuses on the known roles of MAPK/ERK signaling in renal differentiation and discusses the missing pieces that are needed to complement our understanding of this important topic.

### 1.1. RTK Signaling in General

The RTKs form a family of cell-surface receptors possessing an extracellular domain that binds to signaling molecules such as growth factors, hormones, cytokines, and neurotrophic factors [10]. RTK signaling has important functions in regulating cellular processes such as cell cycle progression during proliferation, cell metabolism, migration, survival, and differentiation. Several growth factors (e.g., epidermal growth factor, insulin, and the insulin-like growth factor) bind to RTKs, upon which the tyrosine residues within the receptors are phosphorylated, leading to changes in either downstream target gene expression and/or cytosolic protein phosphorylation [11]. The intracellular pathways mediating extracellular stimuli into the cell interior include, for example, PI3K/AKT/mTOR, RAS/RAF/MEK/ERK, and PLCγ/PKC [12,13] (Figure 1).

### 1.2. MAPK Pathway

Four distinct MAPK pathways execute RTK functions through the small G-protein RAS (Figure 1). The best characterized is the RAS/RAF/MEK/ERK pathway, which in mammals constitutes three RAF proteins (RAF1, A- and B-RAF), two MEK (MEK1 and -2), and two ERK (ERK1 and -2) proteins. Activated RAS generally stimulates RAF but may also phosphorylate MEKK2/3, thus inducing activation of another MAPK cascade consisting of MEKK2/3-MEK5-ERK5 [14]. Analogous to other MAPK pathways, sequential protein kinase phosphorylation via Rho-MEKK1(or MLKs)-MKK4/7 activates JNK1/2/3. Yet another parallel MAPK pathway consists of RAC (or Cdc42)-TAK1 (or MTK1)-MKK3/6 proteins to finally activate the effector protein p38 [15,16,17,18,19,20,21]. Highlighting the complex nature of cascade activation, RAS proteins not only induce the MAPK/ERK pathway but also activate the PI3K pathway [22,23]. The regulation and molecular events that determine which intracellular cascade gets activated upon RTK signal transduction are not fully known. Interestingly, high levels of ERK and p38 were detected by Western blotting and immunostaining in the rat embryonic kidney, while JNK was abundant in the adult kidney [24]. Moreover, in vitro experiments also suggest involvement of the JNK pathway during embryonic kidney development [25,26]. These observations imply a functional requirement of MAPK/ERK, p38, and JNK pathways for renal differentiation [27,28].

#### 1.2.1. MAPK/ERK Cascade

The first identified MAPK pathway, the ERK1/2 cascade, is activated by various stimuli and contributes to the regulation of proliferation, differentiation, cell survival, learning, migration, apoptosis, morphology determination, and oncogenic transformation [29]. After RAS engagement, RAF protein kinases RAF1, B-RAF, and A-RAF are activated and phosphorylate their dual-specificity protein kinase substrates MEK1 and MEK2 (also known as MAP2K1 and MAP2K2) [30]. Subsequently, MEK1/2 phosphorylate their substrates, the best-known of which are ERK1/2, which have a wide range of both cytosolic and nuclear targets and functions [30,31].

Interestingly, a significant portion (50–70%) of ERK1/2 molecules can be found in the nucleus within 10–20 min of cellular stimulation, and this appears to be a key step for the quick regulation of cell cycle progression and DNA replication [32,33]. Recent work by Johnson and Toettcher demonstrated that not only signal duration but also total amplitude of ERK activity specifies the embryonic lineages in *Drosophila melanogaster* [34]. By combining optogenetic stimulation and genetic inactivation of ERK, they found that ectodermal cell types are induced with a fast ERK pulse, while differentiation of gut endoderm requires a longer ERK stimulus. Finally, they showed that incremental ERK activation leads to a switch in cell fates during endoderm/ectoderm specification thus indicating that total ERK load rather than amplitude or duration of ERK activity controls fate decisions. Other reports show that ERK signaling regulates pluripotency in mouse and human embryonic stem cells (ESCs). In ESCs, the MAPK/ERK pathway controls genomic stability, self-renewal, and differentiation [35,36]. Further highlighting the importance of the MAPK/ERK pathway in tissue homeostasis, mutations in *RAS*, *B-RAF*, *RAF1*, *MEK1*, or *MEK2* appear to drive uncontrolled cell proliferation and survival of cancer cells in a growth factor-independent manner [37,38]. Mutations in the negative regulators of MAPK/ERK pathway activation (such as *NF1*) significantly contribute to tumor biology and emphasize the importance of pathway control at multiple levels [39,40].

#### 1.2.2. Other MAPK Cascades

Unlike the ERK pathway, which typically mediates RTK signaling, JNK and p38 pathways respond to a variety of stress signals [41]. For example, JNK and p38 pathways are activated by proinflammatory cytokines, such as tumor necrosis factor (TNF)-α and interleukin (IL)-1β, and by cellular stresses [42]. JNK is involved in apoptosis, immune cell development, and autophagy [43,44]; while p38 can additionally control cell cycle, growth, and differentiation [45]. There is evidence that both JNK and p38 pathways may participate in kidney development through the control of nephron differentiation [25,26].

Considerably less is known about ERK5 than the other MAPK pathways. It is activated by inflammatory cytokines, ischemia and hypoxia, and to a lesser extent by RTK signaling. The ERK5 pathway is linked to diverse cellular processes including proliferation, migration, survival, and angiogenesis [46]. In general, ERK5 is universally expressed in many tissues, but the highest level is detected in the brain with peak levels during early embryonic development [47]. The role of the MAPK/ERK5 pathway in kidney development has not been examined.

### 1.3. MAPK/ERK Pathway: Lessons from Mouse Models

Genetic experiments targeting multiple components of the RAF/MEK/ERK pathway have greatly facilitated the understanding of the consequences of MAPK/ERK activation in vivo. Typically, constitutive loss-of-function of MAPK/ERK pathway components (e.g., *B-raf*, *Raf-1*, *Erk2*) result in early embryonic lethality [48,49,50,51,52]. Therefore, gain-of-function mouse models together with conditional alleles are essential for the characterization of the tissue- and organ-specific requirements for MAPK/ERK activation [53,54,55]. Since this review focuses on renal differentiation, we will discuss in detail the genetic models of *Mek1* and -*2*, which have been utilized in studies of the developing kidney and compare those findings to what has been discovered in *Erk1* and -*2* models.

Genetic inactivation of *Mek1* in mice demonstrated its indispensable function in embryogenesis [56]. Characterization of *Mek1* mutant mice revealed that the embryonic lethality is caused by defects in the placenta, which exhibited decreased amounts of endothelial cells in the labyrinthine region. Interestingly, the tetraploid complementation of mutant embryos rescues the placental phenotype and results in viable offspring, suggesting that MEK1 function is not required for the development of the embryo proper [57]. Also, *Mek2* deletion supports normal embryogenesis and results in viable offspring [58]. However, double heterozygosity of *Mek1* and *-2* results in a phenotype very reminiscent of that observed in *Mek1*-null embryos, thus demonstrating that while MEK1 functions appear to predominate, both kinases are essential for normal placental development [59,60]. The work by Aoidi and colleagues further supports the redundant function of MEK1 and -2, which shows that genetic replacement of *Mek1* with *Mek2* under the control of endogenous *Mek1* regulatory elements fully supports normal development [61]. These experiments also clearly demonstrate an essential effect of gene dosage for functional ERK signaling.

It has additionally been shown that MEK1 can negatively regulate MEK2-dependent activation of ERK signaling in cultured cells and MEK1-deficient embryos [62]. This suggests that MEK1 and -2 are not interchangeable, and as shown, loss of MEK1 may increase ERK1/2 phosphorylation. Despite this, combinational deletion of MEK1 alone or together with MEK2 has shown that MAPK signaling participates in the regulation of synaptic plasticity and development of excitatory neurons in the neocortex, functional skin barrier, and the respiratory tract [5,7,63,64].

Similar to MEK mouse models, genetic inactivation of *Erk2*, but not *Erk1*, results in embryonic lethality due to placental defects [52,65,66,67]. Although ERK1-deficient mice appear normal, ERK1 has been implicated in asthma (due to its requirement for T-helper cell differentiation), resistance to obesity and skin papillomas, long-term memory function, and control of hyperactivity [68,69,70,71,72]. Because homozygous deletion of *Erk2* in mice is embryonic lethal, hemizygote disruption (*Erk^−/+^*) and hypomorphic mice have been utilized to reveal its functional requirements for development and organ functionality (e.g., for normal long-term memory [73]).

Tissue-specific roles of ERK1 and -2 and their potential physiologically distinct functions have been under intensive research during the last decade. Concluding from extensive experiments performed in both in vitro cell culture and in vivo mouse models, it appears that ERK1 and -2, again similar to their upstream kinases MEK1 and -2, have redundant functions [74,75]. Misinterpretation of some of these results may derive from the differences in expression levels of ERK1 and -2, as the latter is more abundantly expressed in the majority of tissues [75]. When this is taken into account, both proteins are equally important (e.g., for normal proliferation [76]). This is also supported by genetic experiments that have revealed the essential involvement of ERK1 and -2 in craniofacial development, neurogenesis, hematopoetic stem cell maintenance, and placental differentiation [8,77,78,79,80].

## 2. Kidney Development

Kidneys develop via branching morphogenesis, similar to many other organs including the lungs, mammary glands, and seminal vesicles. Novel branch generation specific to each organ determines the characteristic growth and shape of these organs. As such, each organ has its own unique pattern of epithelial branching that is determined by signals from the surrounding nascent mesenchyme [81,82].

Mammalian renal development occurs in three spatially and temporally distinct stages: The pronephros, mesonephros, and metanephros. All three stages derive from the intermediate mesoderm [83,84]. On embryonic day 8.75 (E8.75) in mice (gestational day 28 in humans [85]), paired nephric ducts emerge and grow posteriorly until they contact and connect to the cloaca. During this time, the nephric ducts also induce the growth of the pro- and mesonephric tubules [86,87,88], which are transient structures in females. In males, the caudal mesonephric tubules and distal nephric duct contribute to the differentiation of the accessory organs of the reproductive system [89] (Figure 2).

### 2.1. Kidney Induction—Formation of the Ureteric Bud and Branching Morphogenesis

The adult kidney, or metanephros, begins forming at E10.5 in mice and during weeks 4–5 of gestation in humans, when a mass of metanephric mesenchyme (MM) cells induces UB outgrowth from the nephric duct. Several transcriptional regulators are involved with the initial MM formation, including EYA1, SIX1/SIX4, and OSR1 [90,91,92,93,94,95]. Additionally, it has been suggested that PAX2/PAX8 function in guiding the initial MM establishment [93,96,97,98,99]. For a detailed review of kidney development, see Krause et al. [100].

Reciprocal inductive interactions between the MM and the nephric duct guide the initial formation of the UB. These interactions are mediated by secreted molecules and extracellular matrix components, such as GDNF, FGFs, and integrins, which channel molecular instructions to induce cellular responses. For UB outgrowth, MM cells secrete GDNF which binds to the RET tyrosine kinase receptor and co-receptor GFRa1, both expressed in the nephric duct [4,101,102]. All components of the GDNF/RET signaling pathway are required for normal UB outgrowth which takes place by an active movement of the RET^high^ cells within the nephric duct that clusters these cells and leads to UB formation [103,104,105,106,107,108,109,110]. These cellular movements are mediated by ETS transcription factors, ETV4 and ETV5 [111,112]. FGF signaling through receptors FGFR1/2 is likewise important for induction of UB formation, while FGF signaling in the MM, through an unknown mechanism, prevents ectopic UB formation [113,114].

Reciprocal inductive signaling continues to play significant roles in UB branching morphogenesis. Although the importance of GDNF/RET and FGF signaling for UB branching has been demonstrated in detail [3,104,115,116], downstream targets mediating their functions are still largely unclear. Genetically modified mouse models have shown that the balance in GDNF levels essentially dictates the morphology of the primary UB and its subsequent branching as not only reduction and loss of, but also endogenously increased or expanded GDNF, results in severe renal defects [117,118,119,120,121,122]. GDNF activates essential WNT functions that aid the progression of branching morphogenesis [123,124,125,126,127], while strong inhibitory effects of BMP4 on UB branching and GDNF functions are reported [98,128,129,130].

The entire collecting duct system is formed by repeated UB branching [131,132]. The UB is divided into two regions, known as tips and trunks, each having distinct cellular properties. Cells within the UB tips are immature, highly proliferative, and in direct contact with the MM, facilitating their interaction. The tip cells generate new branches and tips [133], and it has been convincingly shown that cells in the UB trunk derive from tip cells that differentiate to form the entire collecting duct system [134,135]. Therefore, tip cells are the progenitors of the entire collecting duct system. Although the specifics of the molecular signatures in UB tip cells and their regulation are largely unknown, RET signaling has been shown to maintain UB cells in the tip niche [110,111,135] (Figure 3).

### 2.2. Nephron Progenitors and Nephrogenesis

Once UB branching begins, the UB in turn reciprocally induces the condensation of the MM around the tips. This condensed mesenchyme surrounding each UB tip is referred to as the cap mesenchyme (CM) and acts as a source of nephron progenitors (NPs) [136]. Molecularly, NPs are defined by the expression of transcription factor SIX2 and transcriptional cofactor CITED1 [137,138]. NPs form their own niche, which is arranged by contacts between NP cells as well as contacts with the UB and stromal progenitors, but the mechanisms behind the niche arrangement are only about to be elucidated [9,139].

The multipotent, self-renewing NP population differentiates to give rise to the functional units of the kidney, the nephrons. Each newly formed UB tip induces the differentiation in subsets of NP cells which first cluster under the UB tip to form a pretubular aggregate. As nephrogenesis proceeds, the pretubular aggregate epithelializes to form the renal vesicle (RV), which grows and patterns into a comma-shaped and then S-shaped body. The distal S-shaped body differentiates to become the distal and connecting tubule, which plumb into the collecting duct system allowing fluid flow from the kidney [140]. The proximal and medial segments of the S-shaped body give rise to the glomerulus and proximal tubule/loop of Henle, respectively [81,141]. NP cells are gradually recruited into the epithelial nephron precursors, and the timing of the recruitment essentially predicts their future fate in the functional nephron [142].

To assure a sufficient final nephron number, the NP population must be maintained during active UB branching morphogenesis, as indicated by depletion experiments showing that nephron count critically depends on the NP population size [143]. Details of the molecules affecting NP cell commitment were recently reviewed [140]. Nephron progenitor maintenance is controlled by many transcriptional regulators, the most well-known being SIX2, which is required for self-renewal and preventing premature differentiation [144]. Recently, it was also determined that SALL1 has a similar role in NP maintenance as NP-specific deletion of *Sall1* results in premature NP depletion [145,146]. PAX2 maintains and promotes NP cell identity by repressing interstitial programs [147]. Further evidence suggests that PAX2 also specifies the initial kidney progenitor populations in the intermediate mesoderm [148] and participates in the regulation of NP differentiation [96]. Interestingly, PAX2 may have different functions in humans, as suggested by recent experiments in PAX2-deficient iPS cells where the cells were competent to epithelialize into mature nephron cell types [149]. Other noteworthy regulators of NP maintenance include p53, OSR1, EYA1, and N- and c-MYC [150,151,152,153].

Additionally, growth factor signaling via FGF and bone morphogenetic protein (BMP) maintains nephron progenitor cells as a self-renewing population, and likely acts as an upstream regulator for the above-mentioned transcription factors, though the specifics of such regulation are only emerging. FGF signaling has additional functions in nephrogenesis, where it acts together with WNT and Notch signaling to induce the timely differentiation of progenitor cells into functional nephron segments [154,155,156,157,158,159,160,161,162,163,164,165,166,167]. Thus, FGF-induced signaling expands the NP population [168,169], which is especially relevant for experiments aiming to develop in vitro tools and methods mimicking in vivo kidney development [170,171].

Rapid UB branching morphogenesis continues until mid-gestation and slows down after E15.5 [131,132]. Nephron progenitors, however, are maintained until postnatal day three in mice by an unknown mechanism that is no longer directly linked to UB branching [132,172,173]. It is clear that nephrogenesis ceases when the NP cells lose their capacity for self-renewal and instead undergo accelerated differentiation during the final postnatal burst of nephrogenesis. While the exact mechanisms of nephrogenesis cessation remains unclear, it is likely linked to metabolic changes and increased blood flow and oxygen in the niche during this time [174,175]. Elevated mTOR and reduced FGF20 could also contribute to the exit of progenitors from the NP niche [157] while HAMARTIN (encoded by the *Tsc1* gene) was recently identified as a possible regulator of nephron progenitor aging [176].

## 3. MAPK/ERK Signaling Guides Ureteric Bud Branching

Our own work and that reported by others shows that several cell types of the developing kidney rely on MAPK signaling [9,27,177,178]. Experiments with in vitro kidney cultures originally pointed out the functional importance of MAPK signaling for the nephron numbers and UB branching morphogenesis by showing that inhibition of the MEK proteins led to a decrease in proliferation in the UB tip cells and abnormal branch formation [27,133,177].

During UB outgrowth, MAPK/ERK activity is polarized to the side of the nephric duct, giving rise to the UB, while ERK activity is completely lost in RET knockout ducts [110]. After the formation of the initial UB, prominent but heterogenous MAPK/ERK activation is detected both in the epithelium and the surrounding MM. Upon initiation of branching morphogenesis, the signaling gets restricted to UB tip regions [9,178]. Perhaps reflecting the heterogenity in the UB tips, ERK activation strength, when measured as fluorescent intensity in Förster resonance energy transfer (FRET)-based biosensor for ERK, is slightly lower than that detected in nephron progenitors and precursors. The UB trunk shows clearly the weakest activation in developing kidney.

The importance of the MAPK pathway for UB branching morphogenesis was first genetically recognized by disruption of the RET docking site (Y1062), which is involved in the downstream activation of MAPK and PI3K [116,179,180,181,182]. These experiments suggested that MAPK and PI3K activation downstream of RET is essential for kidney growth and patterning through UB branching and nephric duct connection to cloaca [4,183]. However, these experiments did not specifically address the functional requirement for MAPK/ERK signaling, and thus its role in collecting duct morphogenesis remained elusive.

Genetic experiments addressing UB-specific MAPK/ERK functions involved conditional inactivation of *Mek1* and conventional deletion of *Mek2*. Loss of *Mek1* alone in the UB resulted in normal renal morphogenesis, similarly to its deletion in the whole embryo and in epidermal keratinocytes [7,57]. Also, reduction of the *Mek1/2* gene dosages (double heterozygocity and ¾ loss of alleles) in any combinations supported normal UB branching and renal differentiation, thus suggesting a high redundancy between these protein functions in the developing kidney as well. Importantly, deletion of both *Mek1* alleles in a *Mek2* knockout background resulted in normal UB outgrowth but demonstrated a requirement of MAPK signaling for UB branch formation. Abrogation of MAPK activity in the UB results in a major decrease in novel branch formation as well as an elongation-only phenotype [178].

Characterization of the cellular and molecular causes of the branching morphogenesis defect in MAPK-deficient UBs revealed two essential processes: The first is not so surprisingly involved in regulation of cell cycle progression while the second is a clearly less-known function involved in the control of cellular adhesion [178]. The proliferation defect in mutant UBs derives from an inability to increase cyclin D1 levels, which is required for G1-to-S phase progression. Subsequently, less UB cells are in mitosis, which causes a significant reduction in the total number of UB epithelial cells. Simultaneous to the cell cycle progression defects, increased and partially mislocalized E-cadherin is detected in plasma membranes of mutant UB cells, suggesting abnormalities in adherens junctions. Moreover, MAPK-deficient UB cells exhibit immature focal adhesions and accumulation of the focal adhesion proteins to adherens junctions [178]. Thus, MAPK/ERK activity does not simply regulate proliferation in the UB to achieve normal branching and growth, but it additionally regulates the dynamic cell adhesions during kidney development. Further studies are needed to determine how MAPK signaling regulates adhesions within UB cells and among the surrounding matrix.

As detected by FRET biosensor for ERK1/2 activation, the MAPK pathway shows very dynamic activation patterns in the UB [9]. This suggests that precise control of signal strength and duration is important in specifying cellular outcomes in the developing kidney. Indeed, excess endogenous GDNF results in an opposite phenotype of that seen with UB-specific loss of MAPK activity as the tips are abnormally expanded and accompanied by short UB trunks. Thus, GDNF positively regulates collecting ductal progenitor cell expansion, and this is mediated by MAPK activity as chemical MEK inhibition rescues both the tip morphology and trunk length [122]. Accurate control of MAPK activation is further supported by ectopic expression of *Sprouty2*, a negative regulator of RTK/MAPK pathway, in the UB, which shows changes in UB branching morphogenesis [184]. Deletion of another family member, *Sprouty1* (*Spry1*), confirmed the fundamental requirement of negative regulation in control of signal strength as *Spry1* knockout embryos exhibit multiple UBs along the nephric duct [117,118]. The roles of negative feedback mechanisms are further demonstrated by the findings that GDNF induces not only *Spry1* but also the expression of two other negative regulators of the MAPK/ERK pathway, *Spred2* and *Dusp6* [112,127]. Strikingly, genetic experiments inactivating *Gdnf* or *Ret* in a *Spry1* knockout background restore the normal UB branching, thus demonstrating how a fine synergistic signaling balance dictates in normal outcome of UB outgrowth and branching [118,185,186]. The exact roles of SPRED2 and DUSP6 in the feedback process remain to be explored.

## 4. The Function of MAPK/ERK in Nephrogenesis

Live-imaging of embryonic kidneys isolated from FRET based ERK biosensor mice revealed that MAPK/ERK activation is amazingly heterogeneous within the NP population and strong in RVs [9]. This suggests that in addition to UB epithelium, the MAPK pathway could participate in the regulation of nephrogenesis. The activation pattern further indicates that the pathway may have multiple functions in the guidance of nephron differentiation. Unlike in the UB where activation of MAPK pathway in a given cell appears to correlate with cell cycle phase and overlaps with expression of G1 and S phase markers [178], no clear link to proliferative or non-proliferative NP cells was found in the MM and differentiating nephrons.

We inactivated MAPK/ERK signaling specifically in the NPs to reveal that MAPK/ERK activity indeed is needed in two distinct processes of nephrogenesis: The first is its essential role in the maintenance of the NP population while the second faithfully adheres to its activation in differentiating nephron precursors and involves permissive functions for progression of nephron differentiation [9]. The renal phenotype of NP-specific MAPK inactivation resembles defects reported for FGF deficiency. Similarity of phenotypes suggests that MAPK/ERK activation mediates the function of multiple FGF ligands (FGF8/9/20) in the maintenance of the NP population, while in differentiating nephron precursors it likely acts downstream of FGF8-induced signaling [154,160,161].

In the NP-specific absence of MAPK activation, the NP niche is disorganized and faces changes in the key molecular mediator of cell-to-matrix adhesion, integrin alpha 8 (ITGA8) [9]. This suggests that MAPK activity in NP cells is essential for their communication with the extracellular matrix. Loss of MAPK activity also causes cell-intrinsic changes, as levels of PAX2 are reduced specifically in mutant NPs. The molecular details of MAPK activity’s regulation of PAX2 remain to be shown, but as suggested by previous experiments [147], we showed that PAX2 is required in NPs to maintain normal ITGA8 [9]. Likewise, FGF-activated FGFR1/2 signaling not only maintains self-renewal of NP cells but also PAX2 expression in this population [113,154]. In conclusion, MAPK activity regulates niche organization and communication with the extracellular matrix.

NP-specific MAPK inactivation results in an almost complete lack of nephrons in newborn mouse pups [9]. This is in line with previous findings where MEK was chemically inhibited in developing kidneys of rat embryos and caused reduced nephrogenesis [27]. In the NP-specific absence of MAPK activity, nephrogenesis proceeds normally up to the RV stage but then halts almost exclusively. Despite the normal-looking initial steps of nephrogenesis, accumulation of an undefined PAX2^+^ nephrogenic mass resembling pretubular aggregates is occasionally observed in kidneys lacking MAPK activity in NPs. As no increase in apoptotic cells specifically in this population nor a similar increase in RV amount was detected, consequences of this phenotype remain to be elucidated [9].

The nephrogenesis arrest at the RV stage correlates with the expression pattern of *Fgf8*, of which deletion also causes a similar nephron differentiation defect [160,161]. Molecularly, MAPK activity in RVs appears to be required for distal expression of WNT signaling mediator LEF1, cell cycle regulator cyclin D1, and NOTCH ligand JAG1. Moreover, mutant RVs fail to proximally upregulate WT1. Altogether, these results suggest that loss of MAPK/ERK activity indeed blocks the progression of nephron differentiation rather than specifies nephron segment identities. Interestingly, although a strong pERK1/2 signal is detected in the connecting segment of nephrons, which fuses with the collecting duct system, no obvious defects were observed in the connections [7]. In summary, MAPK/ERK activation in RVs primes future nephron cells to be receptive for signals from WNT and NOTCH pathways that are needed to propel further differentiation.

## 5. Aberrant ERK Signaling and Wilms’ Tumorigenesis

Abnormal MAPK/ERK signaling is detected in many different types of cancers. The research in this field has focused on adult tissues and identified that the majority of the driver mutations causing malignances in adults are detected in BRAF and RAF, while aberrations in MEK genes are far less frequent [187]. Considerably less is known about the contribution of MAPK/ERK in pediatric tumors, of which leukemia and brain and spinal cord tumors are the most frequent cancers in children. Like neuroblastoma, which affects nerves and originates during development, Wilms’ tumor (WT) is a childhood tumor of kidneys that originates from deficiency in renal differentiation. While there are some indications of aberrant MAPK/ERK activation in renal cell carcinoma [188], due to the focus of this review, we restrict our discussion to MAPK/ERK pathway in pediatric kidney cancer.

WT is thought to arise from undifferentiated MM, which remains in the kidney after birth. The incidence of WT is increased in many syndromes, and mutations leading to WT are genetically heterogeneous [189]. Common mutations causing WT include aberrations of *WT1* expression [190] and biallelic expression of insulin-like growth factor 2 (IGF2) [191]. Recent studies revealed that inactivation of RNA degradation component DIS3L2 and microRNA processing genes increasing PLAG1 expression results in increased IGF2 expression through different mechanisms [192,193]. Thus, an increase in IGF2 appears to act as a common mechanism to maintain undifferentiated MM in postnatal kidneys, but its influence to intracellular pathways requires further characterization. On one hand, increased PLAG1 expression associates with increased mTOR signaling [193], while augmented pERK1/2 as a response to increased IGF2 is observed in a mouse model of Wilm’s tumor [194]. Furthermore, ERK signaling is linked to WT development as upregulation of pERK1/2 occurs in a subset of human tumors [195], but the relevance of activated MAPK/ERK signaling as a driving cause of WT remains to be studied.

## 6. Future Outlook

The prevalence of chronic kidney disease, end-stage renal disease, and acute kidney injury is growing rapidly due to an aging population and an increased incidence of diseases like diabetes. Thus, kidney diseases, whether acquired or congenital, are not only a huge personal agony for patients but also their treatment is an increasing challenge for health care systems worldwide. Despite the clinical demands for the treatment of renal diseases, current therapies, dialysis and transplantation, have limitations in addressing these needs, and thus development of novel strategies to tackle kidney diseases are grievously sought after. High hopes were awoken a few years back when the first kidney organoids were generated from pluripotent stem cells [196,197,198]. The current state of the art, though successful in generation of kidney lineage from pluripotent stem cells, is inefficient in differentiating and propagating collecting duct and nephron progenitors, and thus gives rise to organoids of very small size. Limited knowledge of the molecular and cellular mechanisms regulating expansion of renal progenitor cells and their timely differentiation into functional structures has hampered the progress of taking organoids to the next level. Recent advances in understanding the functions of intracellular pathways, such as MAPK/ERK activation, ought to be next taken into account in organoid differentiation as well as in attempts aiming to reveal genetic causes of congenital kidney diseases. This is especially important in the current situation where organoid differentiation protocols heavily rely on protein supplementation rather than utilization of defined chemical compounds that would allow more precise activation–inactivation control over the given signaling pathway.

Surprisingly few studies have genetically addressed the function of intracellular pathways in developing kidney and thus limited information is available for evaluation of their specific functions. Systematic and tissue-specific inactivation of cascades induced downstream of RTK activation together with the generation of disease-specific mouse models bearing the known disease-causing mutations are the key next steps required for advancing therapeutic interventions in kidney diseases. Therefore, continuous basic research that generates novel information of kidney development together with systematic testing of the nephrogenic and diagnostic potential of these findings is needed for advancing kidney health in the future.

## Figures and Tables

**Figure 1 ijms-20-01779-f001:**
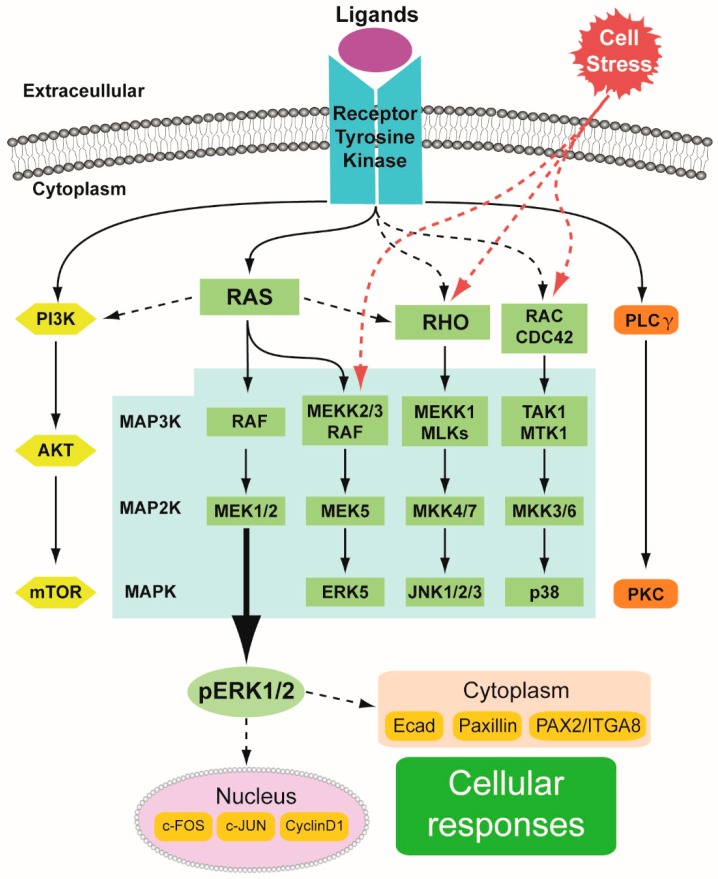
Schematic summary of the multiple MAPK pathways. MAPK pathways, highlighted in the light blue box, mediate extracellular information to the cell interior. Ligand (purple oval) binding to the dimeric transmembrane tyrosine kinase receptor (blue blocks) and/or cellular stress (inflammation, ischemia, hypoxia) activate a spectrum of downstream intracellular cascades such as PI3K/AKT/mTOR, RAS/RAF/MEK/ERK, and PLCγ/PKC. The last step of MAPK/ERK cascade activation results in phosphorylation of ERK1/2 leading to both transcriptional target regulation in the nucleus (light lilac oval, including c-FOS, c-JUN and Cyclin D1) and in the cytoplasm (peach-colored box, including those identified in the developing kidney: E-cadherin, Paxillin, and PAX2/ITGA8) followed by cell-type specific cellular responses (green box). Black lines indicate linear activation by phosphorylation, black dotted line indicates activation within the cascade and red dotted line indicates activation by cellular stress. Abbreviations: PI3K; Phosphoinositide 3-kinases, AKT; protein kinase B, mTOR; mammalian target of rapamycin, MAP3K; MAP kinase kinase kinase, MAP2K; MAP kinase kinase, MAPK; MAP kinase, MLKs; Mixed lineage kinases, MKK; Mitogen-activated protein kinase kinase, CDC42; Cell division control protein 42 homolog, TAK1; Mitogen-activated protein kinase kinase kinase 7, MTK1; Mitogen-activated protein kinase kinase kinase 4, PLCγ; Phospholipase C gamma, PKC; Protein kinase C, pERK1/2; phosphorylated ERK1/2, Ecad; E-cadherin, PAX2; paired box protein 2, ITGA8; integrin alpha 8.

**Figure 2 ijms-20-01779-f002:**
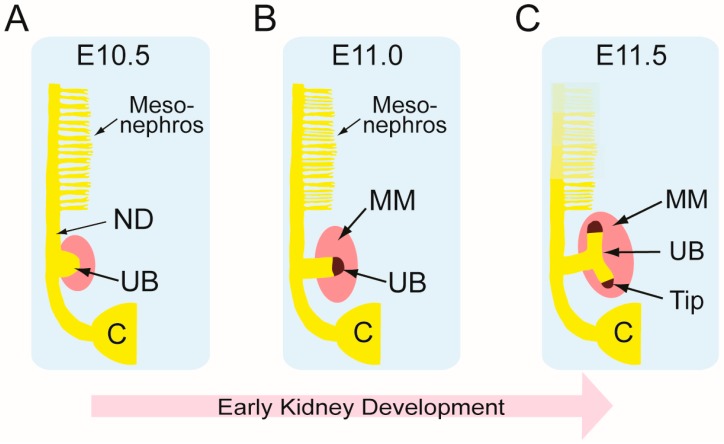
Induction of permanent kidney development in mice. (**A**) The most prominent kidney at embryonic day 10.5 (E10.5) is the mesonephros, which is a transient, embryonic kidney. At the initiation of permanent kidney development, the metanephric mesenchymal (MM, pink) cells interact with the ureteric bud (UB, yellow), and signal propagation between the MM and UB induces UB outgrowth from the nephric duct (ND) near the cloaca (**C**); (**B**) In the next step at E11.0, the MM secretes signals that attract the UB to grow towards it. The UB itself is divided into UB tip (dark brown) and trunk (yellow); (**C**) The first branching event takes place at E11.5 when reciprocal inductive interactions between the UB and MM lead to formation of a T-shaped UB. The previous kidney, mesonephros, begins to regress. C; cloaca, ND; nephric duct, MM; metanephric mesenchyme, UB; ureteric bud.

**Figure 3 ijms-20-01779-f003:**
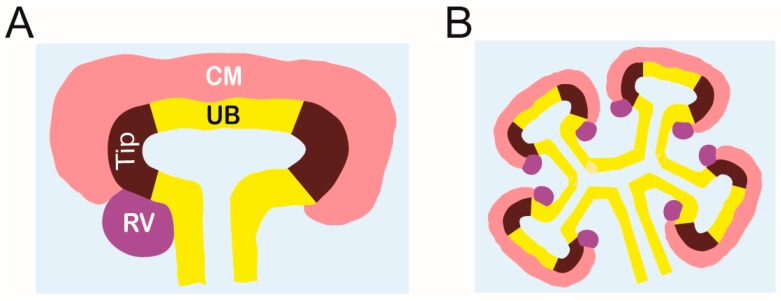
Ureteric bud (UB) branching morphogenesis. (**A**) Branching morphogenesis begins by the formation of a T-bud, which is composed of the UB trunk (yellow) and tips (dark brown). At this stage, the metanephric mesenchyme is condensed around the UB tips and forms cap mesenchyme (CM, light red). UB-derived signals induce CM cells to undergo mesenchyme-to-epithelial (MET) conversion and produce the renal vesicle (RV, purple) in the armpits of the T-bud; (**B**) By E12.5, the UB has undergone a series of iterative branching events under the influence of mesenchymal signals. Most of the UB branches form by stereotypic bifurcation of an existing tip, and each new tip maintains the surrounding CM. The UB undergoes a total of approximately 10 to 11 rounds of branching events during which it induces MET and thus nephrogenesis within each newly generated tip. CM; cap mesenchyme, UB; ureteric bud, Tip; UB tips, RV; renal vesicle.

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
