# Peer review of "MAPK/ERK Signaling in Regulation of Renal Differentiation"

_ijms, 2019, doi:10.3390/ijms20071779_

Round 1
Reviewer 1 Report
In this very informative and timely review Kurtzeborn et al discuss our current understanding of MAPK signaling in kidney development. Whereas multiple ligands of RTKs upstream of MAPK have been known for a long time, surprisingly little work has been done on the exact intracellular pathways they activate in which situation. Yet this is important to couple these extracellular signals that control kidney development to the molecular intracellular events that execute these signals, which in its turn will help to make the link to transcriptional programs, increasingly at single cell level, that are being described by many groups at the moment.
The authors first provide an overview of the different variants of MAPK signaling pathways and the current state of knowledge from the use of knockout mouse models. This overview is clear and very helpful to understand the rest of the manuscript for less-initiated in MAPK signaling. It also makes the point that by far most is known about MAPK/ERK signaling, and the review therefore for the rest focuses on this.
The review continues with a description of kidney development. On one hand this is maybe a bit long and detailed, but on the other hand it is clear that RTKs, and therefore MAPK/ERK are important in many parts of this process. I think that this part and the level of detail will be much appreciated by readers coming from the MPAK/ERK field bul less (or no) background in kidney development.
The roles of MAPK/ERK in UB and nephron biology is discussed in more detail. Again, this is a good overview of the current understanding, and illustrate that a lot more is still to be understood.
Finally, there are short discussions of the potential role of MAPK/ERK Wilms’ tumors. Given the developmental origin of these tumors this is a logical addition, but it makes me wonder if anything is known about MAPK/ERK in other types of renal cancer, both developing/childhood and adult types? Have mutations in this pathway been associated with these cancers, have these pathways been tested for therapeutic intervention?
Overall I thoroughly enjoyed reading this review and have no hesitation in supporting it in principle for publication. The only two minor issues I have are the lack of cellular stresses (mentioned in line 111) from fig. 1, and the use of the name ‘Wingless’ in line 212 (this is only the name of theprotein in fruit fly, not in mammals which are the subject of this review, and this name should therefore not be used here.
Author Response
Reviewer #1
In this very informative and timely review Kurtzeborn et al discuss our current understanding of MAPK signaling in kidney development. Whereas multiple ligands of RTKs upstream of MAPK have been known for a long time, surprisingly little work has been done on the exact intracellular pathways they activate in which situation. Yet this is important to couple these extracellular signals that control kidney development to the molecular intracellular events that execute these signals, which in its turn will help to make the link to transcriptional programs, increasingly at single cell level, that are being described by many groups at the moment.
The authors first provide an overview of the different variants of MAPK signaling pathways and the current state of knowledge from the use of knockout mouse models. This overview is clear and very helpful to understand the rest of the manuscript for less-initiated in MAPK signaling. It also makes the point that by far most is known about MAPK/ERK signaling, and the review therefore for the rest focuses on this.
The review continues with a description of kidney development. On one hand this is maybe a bit long and detailed, but on the other hand it is clear that RTKs, and therefore MAPK/ERK are important in many parts of this process. I think that this part and the level of detail will be much appreciated by readers coming from the MPAK/ERK field bul less (or no) background in kidney development.
The roles of MAPK/ERK in UB and nephron biology is discussed in more detail. Again, this is a good overview of the current understanding, and illustrate that a lot more is still to be understood.
Response:
Thank you for detailing the overall positive feedback and comments from our review manuscript “MAPK/ERK Signaling in Regulation of Renal Differentiation”. We very much appreciate your knowledgeable response and thorough understanding of the field. Please find below our responses to the issues raised by the reviewer.
Finally, there are short discussions of the potential role of MAPK/ERK Wilms’ tumors. Given the developmental origin of these tumors this is a logical addition, but it makes me wonder if anything is known about MAPK/ERK in other types of renal cancer, both developing/childhood and adult types? Have mutations in this pathway been associated with these cancers, have these pathways been tested for therapeutic intervention?
Response:
We agree with the reviewer that the involvement of MAPK/ERK pathway in other types of cancers, whether pediatric or adulthood, is an interesting issue. However, given its well-known contribution to different types of cancers, amazingly little work appears to be done with renal cancers [1]. As an example, a recent review focusing on renal sarcomas does not identify MAPK/ERK pathway involvement in these tumors [2]. Moreover, due to the focus of our review in kidney development, we would rather limit our discussion on pediatric kidney cancer, namely Wilm’s tumor. We have now amended the chapter “Aberrant ERK signaling and Wilms’ tumorigenesis” with above mentioned errands, and added a further publications, which may have relevance with MAPK/ERK activation and Wilm’s tumor (new text with track changes between rows 394-416). Finally, we hope that the reviewer agrees with the restricted discussion on this issue based on the reasoning presented here.
Overall I thoroughly enjoyed reading this review and have no hesitation in supporting it in principle for publication. The only two minor issues I have are the lack of cellular stresses (mentioned in line 111) from fig. 1, and the use of the name ‘Wingless’ in line 212 (this is only the name of theprotein in fruit fly, not in mammals which are the subject of this review, and this name should therefore not be used here.
Response:
We have now modified figure 1 to also include cellular stress as one of the inducers of MAPK pathway, and removed “Wingless” from the lane 214 in the revised manuscript.
1. Oka, H.; Chatani, Y.; Hoshino, R.; Ogawa, O.; Kakehi, Y.; Terachi, T.; Okada, Y.; Kawaichi, M.; Kohno, M.; Yoshida, O., Constitutive activation of mitogen-activated protein (MAP) kinases in human renal cell carcinoma. Cancer Res 1995, 55, (18), 4182-7.
2. Ozturk, H., Prognostic features of renal sarcomas (Review). Oncol Lett 2015, 9, (3), 1034-1038.
Reviewer 2 Report
The authors review how the MAPK/ERK Signaling influences and regulates renal differentiation. The MAPK/ERK pathway is an important pathway involved in nephrogenesis. Understanding of the involved pathways might lead to the development of possible targets for treatments of several kidney diseases and cancers.
Overall the reviews reads well. Therefore, I have only minor comments:
Comments:
1. Figure1: For clarity the figure should be self-explanatory.
The blue blocks should in addtition to be called "Receptors" be also identified as RTK.
Also, for better understanding, all abbreviation used in the legend should appear in the figure and vice versa. Eg. ERK1/2 appears in the legend, but is not in figure. Either the figure or the text should be amended.
2. Line 139/140 Reorder the sentence: The work by Aoidi and colleagues further supports the redundant functions of…which shows that…
3. Chapter 2 in general, in particular Line 193ff The authors, should consider the detailed review on kidney development listing genes important for nephrogenesis is:
Krause M, Rak-Raszewska A, Pietilä I, Quaggin SE, Vainio S. Signaling during Kidney Development. Cells. 2015;4(2):112–132. Published 2015 Apr 10. doi:10.3390/cells4020112
4. Line 350: there is a dot missing at the end of the sentence…regulation of nephrogenesis. The activation…
5. Line 355 to 363: In this paragraph the authors apparently talk about their own research (‘We inactivated…’). However, from the references this is not clear as none of the references refers to their work. The missing citation of the authors’ work needs to be added.
6. Line 376 to line 380: Reference not clear. Is it the one from line 375 or another?
7. Chapter 6: The authors only provide a very generic future outlook, but fail to outline and summarize what exactly are the missing pieces needed to facilitate better understanding of the MAPK/ERK pathway and how they could possibly be tackled in the future.
Author Response
Reviewer #2
The authors review how the MAPK/ERK Signaling influences and regulates renal differentiation. The MAPK/ERK pathway is an important pathway involved in nephrogenesis. Understanding of the involved pathways might lead to the development of possible targets for treatments of several kidney diseases and cancers.
Overall the reviews reads well.
Response:
We want to thank the reviewer for the overall positive feedback and comments from our review manuscript entitled “MAPK/ERK Signaling in Regulation of Renal Differentiation”. Please find below our point-by-point responses to the issues raised by this expert and detailed evaluation.
Therefore, I have only minor comments:
Comments:
1. Figure1: For clarity the figure should be self-explanatory.
The blue blocks should in addtition to be called "Receptors" be also identified as RTK.
Also, for better understanding, all abbreviation used in the legend should appear in the figure and vice versa. Eg. ERK1/2 appears in the legend, but is not in figure. Either the figure or the text should be amended.
Response:
We have now modified figure 1, so that it also includes cellular stress as one mean that activates MAPK/ERK pathway, and changed the used abbreviations to match with figure text and vice versa. Please see the section between rows 50 to 66 in the revised manuscript for details.
2. Line 139/140 Reorder the sentence: The work by Aoidi and colleagues further supports the redundant functions of…which shows that…
Response:
The sentence starting on line 140 pointed out by the reviewer has now been reordered and can be identified by track changes mark.
3. Chapter 2 in general, in particular Line 193ff The authors, should consider the detailed review on kidney development listing genes important for nephrogenesis is:
Krause M, Rak-Raszewska A, Pietilä I, Quaggin SE, Vainio S. Signaling during Kidney Development. Cells. 2015;4(2):112–132. Published 2015 Apr 10. doi:10.3390/cells4020112
Response:
We have now added Krause et al 2015 reference to the suggested location in the manuscript.
4. Line 350: there is a dot missing at the end of the sentence…regulation of nephrogenesis. The activation…
Response:
We have now added period to the end of the sentence indicated by the reviewer, and which is on row 354 in the revised manuscript.
5. Line 355 to 363: In this paragraph the authors apparently talk about their own research (‘We inactivated…’). However, from the references this is not clear as none of the references refers to their work. The missing citation of the authors’ work needs to be added.
Response:
We have now added the missing reference, which can be seen as highlighted in yellow and locates to row 374 in the revised manuscript.
6. Line 376 to line 380: Reference not clear. Is it the one from line 375 or another?
Response:
We have now clarified the references for the indicated chapter, which locates between rows 375 to 383 in the revised manuscript.
7. Chapter 6: The authors only provide a very generic future outlook, but fail to outline and summarize what exactly are the missing pieces needed to facilitate better understanding of the MAPK/ERK pathway and how they could possibly be tackled in the future.
Response:
We appreciate this comment and have now amended the Future overlook chapter in the revised manuscript. It now includes a focused view of missing information, namely what are the tissue specific functional requirements of intracellular cascades in guiding renal differentiation as well as need to better mimic disease causing mutations in animal models, both of which we think fit well to the theme of this review. We hope that the reviewer shares our opinion on this and agrees that the restricted discussion presented in the revised manuscript is enough to lead the reader to the lines of future goals that suit to the topic of our review of literature.